# The Importance of Exploration and Exploitation Innovation in Emerging Economies

António Carrizo Moreira [1,2,3,*], Eurico Colarinho Navaia [1,4] and Cláudia Ribau [2,5]

1 Department of Economics, Management, Industrial Engineering and Tourism, University of Aveiro, 3810-193 Aveiro, Portugal

2 GOVCOPP—Research Unit on Governance, Competitiveness and Public Policies, University of Aveiro, 3810-193 Aveiro, Portugal

3 INESCTEC—Institute for Systems and Computer Engineering, Technology and Science, 4200-465 Porto, Portugal

4 Faculdade de Ciências Sociais e Humanidades, Universidade Zambeze, Beira 369, Sofala, Mozambique

5 Instituto Superior de Contabilidade e Administração, University of Aveiro, 3810-193 Aveiro, Portugal

* Correspondence: amoreira@ua.pt

**Abstract:** Innovation capabilities are among the main driving sources of export performance; however, the literature on how exploration and exploitation innovation influence export performance in the context of emerging economies is scarce. As such, the purpose of this paper is to assess the relationship between innovation capabilities and export performance, mediated by exploration and exploitation innovation. For that, an online questionnaire was implemented surveying 250 Mozambican Small and Medium Enterprises (SMEs) in the agro-industry, agro-processing, and fishing sectors. Based on a Partial Least Squares Structural Equation Model (PLS-SEM) relating innovation capabilities, exploitation, and exploration innovation to export performance, it is possible to state that innovation capabilities influence the export performance of SMEs in this emerging country. Moreover, exploration and exploitation innovation exert important mediation effects, the latter being more prevalent than the former. The results constitute a major contribution as it is possible to claim that, in the context of emerging economies, Mozambican SMEs have managed to enhance their export performance, based on innovation capabilities, but, also, to master their ambidexterity with exploiting capabilities, normally linked to their current technological trajectories, being more prevalent that exploitation capabilities, which are focused on the pursuit of radical innovation.

**Keywords:** innovation capabilities; exploration innovation; exploitation innovation; internationalization; export performance; Mozambique; SMEs; PLS-SEM

## 1. Introduction

The importance of innovation for firms' and industries' success and competitive survival is not new [1,2]. Several typologies and understandings have been used to analyze innovation and innovation capabilities and how they have an impact on business and/or export performance [3–8].

Regarding the degree of novelty, innovation is linked to two basic results: radical and incremental innovation, in which the latter is associated with proximity to the market or consumer existence and the former is associated with changes that occur closely related with new technological directions [3,9].

Incremental and radical innovations form two different strands within the organization as they are totally different. While incremental innovation is linked to the current technological trajectory, which is normally used by firms seeking to master and enhance their current established organizational capabilities [3,10] so that they can serve their current markets [11,12], radical innovation is normally linked to disruptive technological

changes, so that firms can master new organizational capabilities in order to develop new products and services to serve new customers and transform new markets [10,12].

This paradoxical approach to innovation leads to two different perspectives when analyzing innovation capabilities: exploitation and exploration capabilities that demand different organizational capabilities [13]. Firms that master both capabilities are known for their organizational ambidexterity [14–18].

The importance of (exploration/exploitation) organizational ambidexterity has been analyzed with several different results [14–18]. However, emerging countries have not been addressed: e.g., Yalcinkaya et al. [17] and Arnold et al. [14] focused on firms in the United States, Atuahene [15] analyzed Chinese firms, and Camisón et al. [16] studied the Spanish context.

Emerging economies, contrary to developed economies, are characterized by low per capita income, lack of foreign investors, a fragile regulatory system, and a myriad of companies that lack resources to compete in international markets [19], despite some interesting cases of successfully competing internationally [20,21].

The concept of emerging economy is not new and has been widely used in the economic literature to distinguish groups of developed and undeveloped countries. An emerging economy is defined as an economy that over the past 15 years has demonstrated an average level of GDP per capita lower than the world average and an average GDP per capita growth rate higher than the world average [22]. In these terms, taking into consideration Saccone's classification [22], Mozambique can be considered an emerging economy.

Emerging economies constitute about 42% of the world GDP [23]. However, several studies on innovation have their focus on advanced economies [24]. SMEs in emerging economies can minimize problems by taking advantage of their existing skills and use their knowledge base to improve their export performance [25,26].

Domestic firms in emerging economies face dynamic environments with rapid political, economic, and institutional change. Environmental uncertainties demand firms to upgrade and reconfigure existing resources and competencies to both survive in the short term and create new products and processes to compete in the long term [20]. In addition, Mozambican companies face challenges in terms of business environment, access to markets, access to finance, and coordination of support mechanisms [27,28] to compete in the domestic market, facing international competitors, and also exporting their products to compete in international markets.

In the case of Mozambique, small- and medium-sized firms (SMEs) represent around 96% of the business system, accounting for 42% of local employment and 28% of the gross national product [29]. In this context, innovation capabilities play an important role, driving structural change in developing countries, particularly to achieve international competitiveness [30].

Through exporting, firms can use their internal competencies to improve production efficiency, raising technological quality and service standards within the firm, and increasing their profits and returns. Innovation capabilities, in addition to being important sources of competitive advantage [4,31–33] having an impact on SMEs' export performance [4,6,7], are recognized as key drivers of business growth [34]. However, when one analyzes the geographical origin of the studies, most of them are from European developed economies [6,8,24–26] and from large outward-oriented economies such as Brazil and China [4,7].

While there is empirical research on the determinants of export performance [4,6–8], few authors address the determinants of export performance and the relationship between innovation capabilities and export performance, mediated by exploratory and exploitative innovation [26,35] in developed economies. However, their study in less-favored, emerging economies is non-existent.

As such, in order to extend this important research to emerging economies, and understand the importance of exploitation and exploration innovation capabilities, this paper seeks to address the mediating effect of exploration and exploitation innovation on the relationship between innovation capabilities and export performance.

This paper has theoretical and practical contributions that must be underlined. First, in the context of emerging economies, Mozambican firms demonstrate they were able to take advantage not only of innovation capabilities to enhance their export performance, but also of their ambidextrous exploitation and exploration capabilities to deploy both short-term, incremental innovation and long-term, radical innovation. This is a novelty as SMEs from emerging economies need to invest in both short-term, incremental and long-term, radical innovation activities to manage their way through if they want to successfully compete in international competitive markets.

Secondly, SMEs need to internalize that in order to embrace a more outward competitive perspective they have to invest in innovation skills in order to develop their innovation capabilities in a holistic perspective, integrating production, marketing, and R&D capabilities with learning, organizational, resource exploitation, and strategic perspectives in order to secure a competitive position to generate both short- and long-term innovation strategies in order to outcompete their rivals based on added-value new products and services.

The paper is divided into six sections. After this introduction, Section 2 reviews the relevant literature, including development of the hypotheses, in which we examine the relationships between innovation capabilities and export performance and the mediating effect of exploration and exploitation innovation. The research method is presented in Section 3. Section 4 presents the results and Section 5 the discussion and the most significant implications. Section 6 presents the main conclusions, limitations, and future lines of research.

## 2. Literature Review

### 2.1. Innovation Capabilities

Dynamic capabilities are defined as the ability to integrate, build, and reconfigure internal and external competencies which firms use to deal with dynamic environments [34] and are normally understood as the abilities businesses use to deploy their resources to maximize their business results [36]. Although innovation capabilities (ICs) can be understood as dynamic capabilities, there is a vast array of perspectives addressing innovation capabilities [4–7,37,38], with some contrasting meanings and understandings.

Lawson and Samson [5] understand ICs as mainstream and newstream innovation, in which the former are regarded as the basic capabilities for firms to succeed and are normally related to mainstream organizational processes to reinforce efficiency and short-term oriented profitability. Manufacturing and marketing capabilities are examples of mainstream innovation capabilities. Newstream innovation capabilities are related to those aspects that need a long-term perspective of the firm's future, which forces managers to face the future embracing change, but minimizing instability.

ICs are also understood as incremental and radical ICs [3,10,39] and are normally related to a short- and long-term orientation so that firms can adjust their capabilities and cope with stable and disruptive technologies and markets. Djoumessi et al. [37] put forward three different types of ICs: institutionalizing, implementing, and stimulating ICs. Institutionalizing innovation seeks to articulate the firm's vision and strategy in tune with newstream ICs, proposed by Lawson and Samson [5]. Implementing ICs is linked with the way firms communicate and share knowledge, and invest in the required resources so that employees have adequate training to perform their duties and fulfil their responsibilities. Finally, stimulating ICs include reward systems to promote creativity to enhance innovation among employees so that difficult challenges can be achieved and new business routines implemented.

One of the most cited papers on ICs is Guan and Ma's [4]. They characterize ICs in seven different dimensions that support firms in successfully transforming knowledge and ideas into innovative processes, products, services, and systems. Based on Teece et al. [34], Guan and Ma [4] put forward seven different capabilities: research and development (R&D) capability; manufacturing capability; marketing capability; resource exploiting capability; organizational capability; learning capability; and strategic capability.

R&D capabilities are related to firms' capacity to invest in R&D activities so that they can adopt unique approaches when developing new technologically-endowed products or processes. Those R&D capabilities include employing qualified industrial experts and acquiring new technologies [2,4,40]. R&D capabilities are mandatory so that firms can develop new products or processes and are related to what Lawson and Samson [5] call mainstream capabilities. Manufacturing capabilities refer to firms' capacity to deal with the diligences of the shop floor and manufacturing activities to transform R&D results into products, according to market wishes and needs. The results regarding the influence of manufacturing capabilities are contradictory, as Guan and Ma [4] and Yam et al. [41] conclude that manufacturing capabilities do not influence performance, whereas Ribau et al. [26] contend that manufacturing capabilities influence export performance. Marketing capabilities are related to the ability to target, position, and segment specific markets. As such, understanding consumers' current and future needs and desires are important to address their purchasing habits and differentiate products and services accordingly. The results regarding the influence of marketing capabilities are also contradictory, as Guan and Ma [4] found that marketing capabilities influence performance, whereas Ribau et al. [26] found that marketing capabilities do not influence export performance.

Firms are complex entities composed of individuals grouped in functional or business units in well-defined organizational structures. The coordination of activities and individuals, according to the business vision and strategy, can only be achieved with shared objectives. As such, if innovation is to happen, innovative projects need to be coordinated among technical (e.g., engineering, manufacturing, and R&D projects), sales, marketing, and manufacturing departments so that innovative products and services can be the result of innovation processes. To this end, flexible innovation processes need to be drawn up so that management techniques can improve work practices and routines to facilitate the share of information and knowledge and enhance the skills within the organization. Thus, organizational capabilities are the enablers that deploy the implementation and distribution of responsibilities so that decision-makers across the organization can make innovation flourish [42].

Learning capabilities are closely linked with the capacity to identify, assimilate, and exploit new knowledge and harmonize resources and capabilities throughout the organization [4,40]. Learning capabilities, as an IC dimension, support the coordination and speed of innovative processes and underpin knowledge-sharing and transfer across the entire organization. Furthermore, learning capabilities are also important in identifying new skills or technologies to support the development of new products and services to embrace new markets [4].

Resource-exploiting capabilities are related to the organizations' capacity to generate, mobilize, and manage their financial, human, organizational, intellectual, and technological resources. Those resource-exploiting capabilities need to combine two different perspectives: generating a flow of financial and managerial resources so that organizations are able: (a) to compete in the market with their current portfolio of products and technologies; and (b) to generate new technologies to develop a new portfolio of products for new markets. These resource-exploiting capabilities are important to generate product and process innovation and need to be tuned to organizational and learning capabilities [4,40].

Strategic capabilities are related to organizations' capacity: (a) to embrace long-term perspectives underpinned by innovative activities and actions; and (b) to be flexible enough to set new directions whenever necessary in order to cope with new goals and business contexts [4]. Those strategic capabilities are important not only to set new strategic directions but also to fine-tune organizational and learning capabilities to mobilize resources so that the organization can cope with the business environment.

There are several perspectives of innovation capabilities [43]. Their constructs are multifaceted [4–7] due to different methodological and company-specific characteristics, the metrics used, and contexts [8,44].

Likewise, a diversity of concepts about ICs can be found. For example, Lawson and Samson [5] define ICs as abilities to continuously transform knowledge and innovative ideas into new products, production processes, and systems for the benefit of the firm and stakeholders. For Guan and Ma [4], ICs are a firm's assets related to internal and acquired experiences. Akman and Yilmaz [45] define ICs as organizational culture, promotional activities, and abilities to perceive and cope appropriately with the external environment. Hogan et al. [46] and Saunila [1] use Lawson and Samson's [5] approach as the basis of their concepts. Although the concept presented by Lawson and Samson [5] is more comprehensive and the most cited in Google Scholar, we will consider in this article the scales presented by Guan and Ma [4], which are the most cited on the Scopus Database, for mirroring clarity and being able to fit Mozambican SMEs.

Export performance has been measured in various ways [47,48], and there is no single, generally accepted metric for measuring it. For example, it can be measured using financial indicators—such as export sales and profits—and non-financial indicators—which include some strategy-based items, such as export goals, satisfaction, and perceived success [6,48].

Several studies support a positive relationship between ICs and export performance [4,6,7]. Guan and Ma [4] analyze how ICs influence the export growth of 213 Chinese manufacturing firms. Oura et al. [7] analyze a sample of 112 Brazilian industrial SMEs and confirm that ICs influence export performance. Analyzing 147 Portuguese SMEs in the plastics industry in Portugal, Ribau et al. [6] conclude that ICs positively support export performance. Those three studies addressed innovation capabilities based on the seven dimensions of Guan and Ma's [4] ICs (R&D, manufacturing, marketing, resources exploitation, organizational, learning, and strategic capabilities). Vicente et al. [8] also claim that ICs have a positive impact on export performance, although innovation capabilities were based on four dimensions (product development capability, strategic capability, innovativeness, and technological capability). As such, it is possible to present the following hypothesis:

**Hypothesis 1 (H1).** *Innovation capabilities have a direct positive effect on SMEs' export performance.*

### 2.2. Exploration and Exploitation Innovations

The term exploitation innovation refers to routine behaviors involved in refining firms' current innovation capabilities, which are associated with improving the performance of existing routines [26,49]. In turn, exploration innovation seeks to respond to latent environmental trends by creating innovative technologies or products that contrast with firms' current technologies in order to serve new markets [26,49]. Furthermore, exploration and exploitation innovations constitute a set of processes that use specific inputs that are transformed into innovative outcomes [50,51]. In distinguishing one from the other, exploitation innovation refers to the search for new knowledge and skills in developing and improving existing products, usually associated with markets where the company is already present. Exploration innovation, on the other hand, uses existing knowledge and skills directed to the development of unique products in order to cope with changing environments, where companies seek to adopt risky innovative strategies to address future markets associated with radical innovations [52]. This perspective is in line with Eisenhardt and Martin [53] and Teece et al. [34], who consider that exploration and exploitation innovations are part of firms' dynamic capabilities.

Exploration capabilities are normally long-term-oriented and are related to the adoption of new products, services, and processes that are new to the firms and have not been used in the past. Differently, exploitation capabilities are used to improve continuously firms' existing resources and processes [17].

Exploration capabilities need to be nurtured by innovative behavior where firms seek to deploy innovative risk-taking behavior, based on R&D activities, creativity, discovery, and experimentation. Complementarily, exploitation capabilities are characterized by incremental, routine-based behavior based on disciplined problem-solving orientation seeking to increase efficiency [40,54].

As referred to above, ICs comprise several different capabilities [4] that characterize firms' core activities, such as R&D, manufacturing and marketing capabilities, and supplementary activities that rely on resource exploiting, organizational, and learning and strategic capabilities. ICs support incremental and radical innovation activities, based on the short- and long-term perspectives of the firm's innovation activities.

Exploitation and exploration innovations increase companies' capabilities to obtain and sustain competitive advantages, and these capabilities are directly influenced by the unique skills and competences of each company [52,55], with positive effects on SMEs' performance [26,50]. Lisboa et al. [56] analyze the relationship between both export market exploitation and export market exploration on export performance and conclude that export market exploitation is positively related to export performance, while export market exploration is negatively related. Also in Portugal, analyzing the plastics industry, Ribau et al. [26] conclude that both exploitation and exploration capabilities positively influence export performance. When examining the antecedents of firms characterized by high levels of exploitation and exploration innovation, Jensen et al. [57] conclude that firms operating in dynamically competitive environments seek both types of innovation simultaneously. Mueller et al. [51] conclude that national culture has a strong impact on the success of exploitation innovation, while only the prevention of uncertainty influences the benefits derived from exploration innovation. As such, even in emerging economies, based on previous studies on ICs and ambidexterity [26,50,56,57], it is possible to support the following two hypotheses when firms are really committed to embrace international competitive markets as they need to be tuned to short- and long-term innovation activities:

**Hypothesis 2 (H2).** *ICs have a positive effect on exploration innovation activities.*

**Hypothesis 3 (H3):** *ICs have a positive effect on exploitation innovation activities.*

Exploration innovation activities seek to take advantage of existing knowledge and skills so that firms deploy their resources to embrace a risky behavior and to develop products and services to embrace new market needs [26,49]. In the case of emerging countries, SMEs need to overcome their short-term perspective and try to serve new products that are not normally developed and sold in domestic markets. As such, only firms with strong dynamic capabilities are capable of succeeding in international markets. In this case, it is expected that internationalized SMEs would be willing to assume the risk of serving new needs that are not normally present in their domestic market. As such, it is possible to hypothesize that:

**Hypothesis 4 (H4).** *Exploration innovation activities have a positive effect on export performance.*

Exploitation innovation is related to incremental innovation, i.e., routine behaviors focused on improving the firm's current ICs [26,49] in order to improve performance. Exploitation innovations seek new knowledge and skills so that firms develop and improve their existing products in the markets they currently serve [43,44]. As such, it is expected that Mozambican SMEs seek to improve their performance by improving their current knowledge and skills so that they gain a competitive edge through performance-enhancing activities to gain competitive edge in the short term. Thus, it is possible to pose the following hypothesis:

**Hypothesis 5 (H5).** *Exploitation innovation activities have a positive effect on export performance.*

Exploration and exploitation innovation capabilities are important elements influencing firms' short- and long-term orientations and are expected to influence the relationship between firms' internal capabilities and export performance. The results indicate a positive influence of exploration innovation capability on SMEs' export performance, in contrast to exploitation innovation capability [26]. For Lisboa et al. [56], the linear, moderate, and

complementary effects of export-market exploitation and exploration are important in explaining export success. However, they argue that export-market exploitation has a positive relationship with export success, whereas export-market exploration is negatively related to export performance [56], resulting from the need for large investments to embrace long-term skills. Thus, despite the fact that Mozambican firms have major resource constraints, an ever-present characteristic among emerging economies, the following hypotheses are proposed, in line with the literature:

**Hypothesis 6a (H6a).** *Exploration innovation positively mediates the relationship between ICs and SMEs' export performance*;

**Hypothesis 6b (H6b).** *Exploitation innovation positively mediates the relationship between ICs and SMEs' export performance.*

Figure 1 shows the proposed conceptual model.

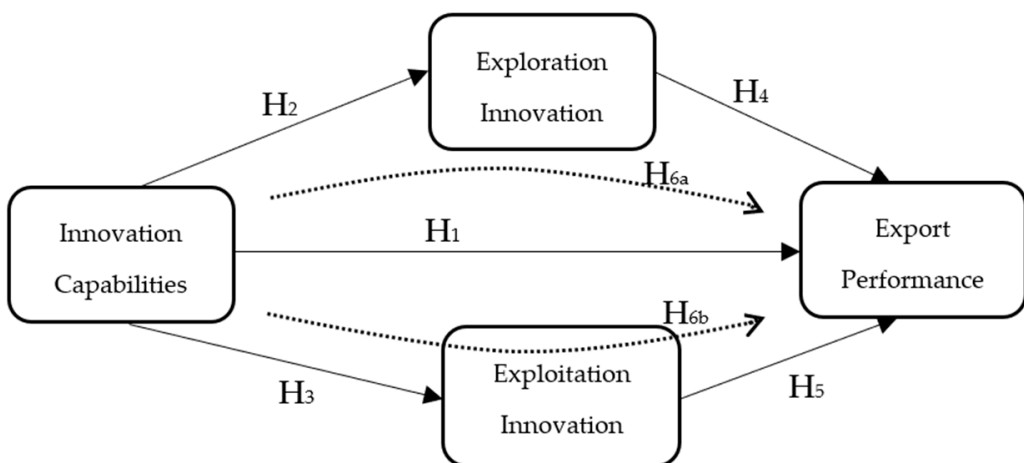

**Figure 1.** Conceptual model.

## 3. Research Method

Data were collected using a questionnaire based on previously adapted and validated scales. Innovation capabilities were measured with a multidimensional scale developed by Guan and Ma [4] and adapted by Ribau et al. [6], namely, learning, manufacturing, R&D, marketing, organizational, resources exploitation, and strategic capabilities. Five items were used for the dimensions of learning, manufacturing, and resources exploitation capabilities; and six items for the remaining dimensions. The constructs on exploration and exploitation innovation capabilities were adapted from Hortinha et al. [35] and Lubatkin et al. [50], with six items for each dimension. Finally, the export performance construct was adapted from previous scales used by Jantunen et al. [58], Kuivalainen et al. [59], Aulakh et al. [60], and Zou and Stan [48]. A seven-point Likert-type scale was implemented, where 1 and 7 serve as anchors, 1 meaning strongly disagree and 7 strongly agree. The main rationale for using these scales is that they have been academically validated. The items for each construct remained unchanged to maintain the original scale.

To achieve the research objectives, the empirical study was based on a non-random convenience sample of managers and owners of Mozambican companies to whom a structured questionnaire was distributed. However, prior to mass distribution, the questionnaire was subjected to a pre-test conducted with a convenience sample of eight individuals (three university professors and five business managers), in order to verify the organization and formatting of the questionnaire, the correct wording, how respondents understood the questions, the response time required, and to ensure the elimination of errors and typos. As a result of the pre-test, some changes were made in terminology to facilitate the respondents'

understanding. The number of items per variable was reduced to a minimum to keep the questionnaire at an adequate size. The final version of the questionnaire was made available online to companies via Google Drive LimeSurvey, for four months. Although this period may be considered excessive, in Mozambique not all companies have the ideal conditions for online information collection, which is why the questionnaire was available for this period.

The Mozambican business population, the target of this study, consists of 43,671 SMEs [29]. After a preliminary analysis of the companies in this database, some inconsistencies were found, with some companies having outdated data and others having already closed down their operations. Thus, based on this universe and with the help of the Investment and Export Promotion Agency (APIEX) of Mozambique, a sample was extracted consisting of 400 exporting SMEs, the target of this study. Following the sending out and receipt of the questionnaires, 305 responses were obtained. After analyzing the collected questionnaires, we excluded 55 due to incomplete responses. In the subsequent analysis, 250 questionnaires with complete responses were analyzed, i.e., 62.5% of the total sample, a very good response rate compared to previous investigations [61,62] and an acceptable number of responses for statistical purposes [63,64]. Although the sample is rather small compared to what is normally desirable when using covariance-based structural equation modeling (CB-SEM), it is still larger than that used in previous research [62,65,66]. This paper complies with the 10-fold rule [64], which indicates that when using PLS-SEM, the sample size should be at least equal to 10 times the largest number of structural paths directed to a particular construct in the structural model.

Statistical analysis of the data was used to test the relationship between the variables analyzed. This analysis was performed using the partial least squares method of structural equation modeling, using PLS-SEM 3.0 software. The use of PLS-SEM can be justified by its robust results on non-normal data and when researchers are focused on predictive models [67]. PLS-SEM produces robust results when the sample is relatively small and the variables are composed of first- and second-order reflective constructs [64,68]. SmartPLS 3.2 was used to run the model proposed.

Regarding the sample of 250 Mozambican firms, 67.2% (168 firms) have fewer than 50 employees, 33% (82 firms) have between 50 and 100 firms, and belong to several industries: 35.6% (89 firms) belong to the wood-processing industry; 26.8% (67 firms) produce and export fishing-related products; and 94 firms (37.6%) belong to the agro-industry and agricultural industries. Regarding respondents, 163 (65%) were the owners of the firms, 79 (32%) were senior managers, and 8 (3%) were functional managers.

## 4. Results

The consistency of the scales was tested following Nunnally [69]. All first-order dimensions have Cronbach's alpha coefficients greater than the minimum threshold value of 0.7, which is a clear indication of good consistency.

This was followed by analysis of the reliability and validity of the measurement. Tables 1 and 2 present information regarding the outer loadings of the first order constructs analyzed. Table 3 presents the path coefficients of the seven ICs forming the innovation capabilities construct. Items with threshold values lower than 0.7 were removed. As such, the results presented in Tables 1–3 support the reliability of the measurement indicators.

**Table 1.** First-order measurement model of innovation capabilities—standardized loadings.

|  | Loadings | *p*-Values |  | *p*-Values | Loadings | *p*-Values |
|---|---|---|---|---|---|---|
| LC1 ← Learning | 0.796 | 0.000 | OC1 ← Organizational | 0.872 | 0.000 |
| LC2 ← Learning | 0.774 | 0.000 | OC2 ← Organizational | 0.859 | 0.000 |
| LC3 ← Learning | 0.839 | 0.000 | OC4 ← Organizational | 0.818 | 0.000 |

**Table 1.** *Cont.*

|  | Loadings | *p*-Values | *p*-Values | Loadings | *p*-Values |
|---|---|---|---|---|---|
| LC5 ← Learning | 0.846 | 0.000 | OC5 ← Organizational | 0.843 | 0.000 |
| MC1 ← Manufacturing | 0.823 | 0.000 | RDC1 ← R&D | 0.765 | 0.000 |
| MC2 ← Manufacturing | 0.865 | 0.000 | RDC2 ← R&D | 0.859 | 0.000 |
| MC3 ← Manufacturing | 0.917 | 0.000 | RDC4 ← R&D | 0.794 | 0.000 |
| MC4 ← Manufacturing | 0.873 | 0.000 | RDC5 ← R&D | 0.879 | 0.000 |
| MC5 ← Manufacturing | 0.840 | 0.000 | REC1 ← Res Explo | 0.923 | 0.000 |
| MKT2 ← Marketing | 0.865 | 0.000 | REC2 ← Res Explo | 0.858 | 0.000 |
| MKT3 ← Marketing | 0.853 | 0.000 | REC4 ← Res Explo | 0.794 | 0.000 |
| MKT4 ← Marketing | 0.897 | 0.000 | SC1 ← Strategy | 0.865 | 0.000 |
| MKT6 ← Marketing | 0.817 | 0.000 | SC2 ← Strategy | 0.951 | 0.000 |
|  |  |  | SC3 ← Strategy | 0.873 | 0.000 |

**Table 2.** First-order measurement model of exploration and exploitation innovation and export performance—standardized loadings.

|  | Loadings | *p*-Values |  | Loadings | *p*-Values |
|---|---|---|---|---|---|
| EI1 ← Exploration | 0.814 | 0.000 | EoiI1 ← Exploitation | 0.825 | 0.000 |
| EI2 ← Exploration | 0.793 | 0.000 | EoiI2 ← Exploitation | 0.805 | 0.000 |
| EI3 ← Exploration | 0.640 | 0.000 | EoiI3 ← Exploitation | 0.880 | 0.000 |
| EI4 ← Exploration | 0.695 | 0.000 | EoiI4 ← Exploitation | 0.871 | 0.000 |
| EI5 ← Exploration | 0.806 | 0.000 |  |  |  |
| EP1 ← Export Perf | 0.887 | 0.000 | EP5 ← Export Perf | 0.867 | 0.000 |
| EP2 ← Export Perf | 0.860 | 0.000 | EP6 ← Export Perf | 0.915 | 0.000 |
| EP4 ← Export Perf | 0.863 | 0.000 | EP7 ← Export Perf | 0.845 | 0.000 |

**Table 3.** Path coefficients of innovation capabilities—standardized path coefficients.

|  | Path Coefficient | *p*-Values |
|---|---|---|
| Learning ← Innov Cap | 0.902 | 0.000 |
| Manufacturing ← Innov Cap | 0.914 | 0.000 |
| Marketing ← Innov Cap | 0.842 | 0.000 |
| Organizational ← Innov Cap | 0.883 | 0.000 |
| R&D ← Innov Cap | 0.811 | 0.000 |
| Res Explo ← Innov Cap | 0.817 | 0.000 |
| Strategy ← Innov Cap | 0.560 | 0.000 |

To complement the reliability analysis of the above model, convergent and discriminant validities were also analyzed. Table 4 presents the values of the average variance extracted (AVE), composite reliability (CR) and correlations of each first-order latent variable. AVE values are greater than 0.50 and all the items have factor loadings of at least 0.70. To ensure convergent validity, internal consistency values, which are expressed by composite reliability (CR), correspond to higher levels of reliability, i.e., CR values are above 0.80, well above the recommended value of 0.60 [70,71]. Discriminant validity was achieved, as shown in Table 4, using the Fornell–Larcker criterion, which argues that the square roots of the AVE values should be greater than the absolute correlation values of the constructs involved. This measure shows that discriminant validity is present in the model [70].

**Table 4.** Discriminant validity.

| Variables | 1. | 2. | 3. | 4. | 5. | 6. | 7. | 8. | 9. | 10. |
|---|---|---|---|---|---|---|---|---|---|---|
| 1. Exploitation innovation | **0.846** | | | | | | | | | |
| 2. Exploration innovation | 0.702 | **0.753** | | | | | | | | |
| 3. Export performance | 0.633 | 0.587 | **0.873** | | | | | | | |
| 4. Learning capability | 0.558 | 0.491 | 0.497 | **0.814** | | | | | | |
| 5. Manufacturing capability | 0.475 | 0.511 | 0.351 | 0.796 | **0.864** | | | | | |
| 6. Marketing capability | 0.447 | 0.547 | 0.389 | 0.634 | 0.793 | **0.859** | | | | |
| 7. Organizational capability | 0.523 | 0.603 | 0.549 | 0.732 | 0.808 | 0.823 | **0.848** | | | |
| 8. R&D capability | 0.293 | 0.262 | 0.328 | 0.775 | 0.628 | 0.546 | 0.558 | **0.825** | | |
| 9. Resources exploitation capability | 0.506 | 0.455 | 0.414 | 0.751 | 0.676 | 0.585 | 0.618 | 0.736 | **0.860** | |
| 10. Strategic capability | 0.097 | 0.293 | 0.233 | 0.505 | 0.366 | 0.307 | 0.405 | 0.611 | 0.416 | **0.897** |
| Cronbach alpha | 0.867 | 0.808 | 0.938 | 0.832 | 0.915 | 0.881 | 0.870 | 0.847 | 0.822 | 0.879 |
| Rho_A | 0.871 | 0.821 | 0.941 | 0.844 | 0.919 | 0.889 | 0.875 | 0.863 | 0.828 | 0.917 |
| Composite Reliability | 0.910 | 0.866 | 0.951 | 0.887 | 0.937 | 0.918 | 0.911 | 0.895 | 0.895 | 0.925 |
| Average Variance Extracted (AVE) | 0.716 | 0.567 | 0.762 | 0.663 | 0.747 | 0.737 | 0.720 | 0.681 | 0.740 | 0.805 |

Note: The values of the diagonal (in bold) are the square root of AVE.

The structural model, in Figure 1, was evaluated taking into account, as suggested by Götz et al. [70], the sign, magnitude, and statistical significance of the parameters of the structural relationships, as well as the coefficient of determination ($R^2$) of the latent endogenous variables. All the structural relationships tested have positive signs and parameters in accordance with the assumptions made when hypothesizing the relationships between the variables. The results are shown in Figure 2 and Table 5.

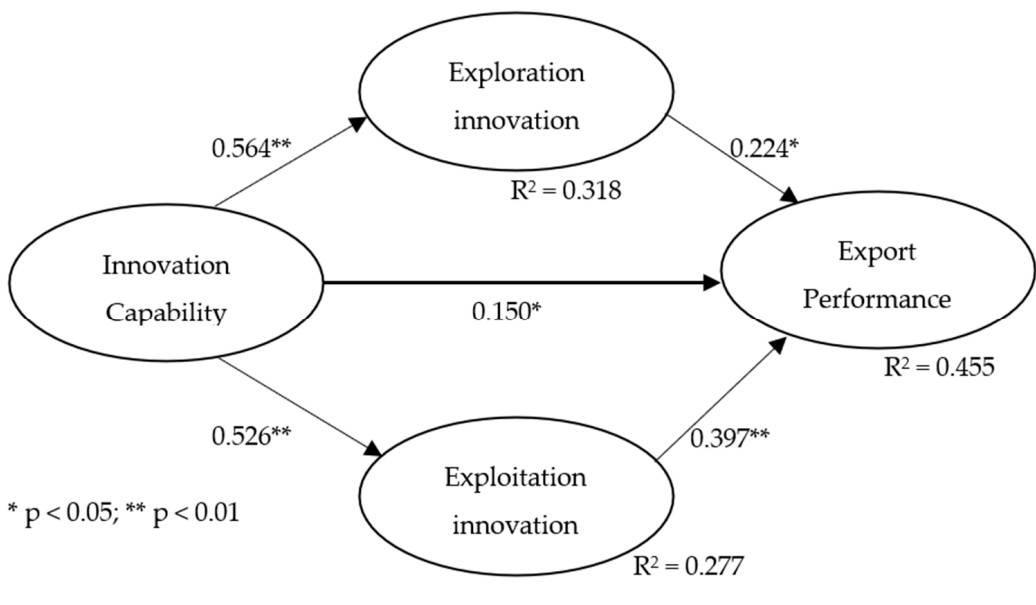

Source: Own elaboration

**Figure 2.** Results of the structural model.

**Table 5.** Direct, indirect and total effects.

| Path | Original Sample | Standard Deviation | T Values | *p*-Values | Hypothesis Validation |
|---|---|---|---|---|---|
| **Direct Effect** | | | | | |
| Innovation capabilities → Export performance (H1) | 0.150 | 0.066 | 2.286 | 0.022 | Accepted |
| Innovation capability → Exploration innovation (H2) | 0.564 | 0.041 | 13.842 | 0.000 | Accepted |
| Innovation capability → Exploitation innovation (H3) | 0.526 | 0.046 | 11.372 | 0.000 | Accepted |
| Exploration innovation → Export performance (H4) | 0.224 | 0.105 | 2.217 | 0.033 | Accepted |
| Exploitation innovation → Export performance (H5) | 0.397 | 0.097 | 4.078 | 0.000 | Accepted |
| **Specific Indirect Effect** | | | | | |
| ICs → Exploitation innovation → Export performance (H6b) | 0.209 | 0.054 | 3.886 | 0.000 | Accepted |
| ICs → Exploration innovation → Export performance (H6a) | 0.126 | 0.057 | 2.199 | 0.028 | Accepted |
| **Total Indirect Effect** | | | | | |
| Innovation capabilities → Export performance | 0.335 | 0.040 | 8.374 | 0.000 | |
| **Total Effect** | | | | | |
| Innovation capabilities → Export performance | 0.500 | 0.037 | 6.823 | 0.000 | |

Source: Own elaboration.

The structural model presented in Figure 2 shows the relationship between innovation capabilities and export performance, being mediated by exploration and exploitation innovation. It is possible to state that 45.5% of the variance of export performance is explained ($R^2 = 0.455$) by its antecedents. Likewise, innovation capabilities explain 27.7% ($R^2 = 0.277$) and 37.8% ($R^2 = 0.318$) of exploitation innovation and exploration innovation, respectively.

Table 5 presents the direct, indirect and total relationships between all variables in the model. All direct relationships between the variables have a statistically significant effect at a 5% significance level. When analyzing the direct effects between the variables, innovation capabilities have a significant direct effect ($\beta = 0.150$) on export performance, which validates hypothesis 1. The relationship between innovation capabilities and exploration innovation is also strongly positive ($\beta = 0.564$) and the relationship between exploration innovation and export performance is positive, although somewhat modest ($\beta = 0.224$), which results in a mediating relationship ($\beta = 0.126$) that complements the direct relationship. When the mediation relationship of exploitation innovation is analyzed, a strong relationship is found between innovation capabilities and exploitation innovation ($\beta = 0.526$) and between exploitation innovation and export performance ($\beta = 0.397$). Thus, it can be concluded that the mediation relationship of exploitation innovation ($\beta = 0.209$; t = 3.886; and $p < 0.000$) is stronger than that of exploration innovation ($\beta = 0.126$; t = 2.199; and $p < 0.028$). It is therefore possible to validate the research hypotheses H2, H3, H4, H5, H6a, and H6b.

In order to analyze the strength of the mediation effect, we used the three-factor approach proposed by Zhao et al. [72], which can determine how much of the indirect effect absorbs the direct effect. For this variance accounted for (VAF) it is calculated how much of the direct path is absorbed. Based on the value of VAF, the following mediation effect conditions are given by Hair et al. [63]:

- No mediation if $0 < \text{VAF} < 0.20$;
- Partial mediation of $0.20 < \text{VAF} < 0.80$;
- Full mediation if $\text{VAF} > 0.80$.

Given the values presented in Table 5, the VAF value of exploration innovation—$(0.126)/(0.126 + 0.150) = 0.457$—indicates that it partially mediates the relationship between innovation capabilities and export performance. The same is true for exploitation innovation capabilities—$(0.209)/(0.209 + 0.150) = 0.582$. Finally, the combined effect of exploration and exploitation innovations—$(0.335)/(0.335 + 0.150) = 0.691$—partially mediates the relationship between innovation capabilities and export performance.

## 5. Discussion

The results in Table 5 show that the direct effect between ICs and export performance ($\beta = 0.150$; t = 2.286; and $p < 0.022$) is statistically significant, validating H1 and confirming previous literature on firms competing in different countries [4,6,8]. Although most Mozambican SMEs, such as other firms in developing countries, particularly in sub-Saharan Africa, have structural problems, low levels of productivity, limited modern technology, and, as a consequence, low product quality [30], the few that compete abroad have managed to follow a growth strategy and been able to leverage their ICs to influence their export performance positively.

ICs also contribute positively to both exploitation and exploration innovation ($\beta = 0.564$; $p < 0.001$ and $\beta = 0.526$; and $p < 0.001$, respectively). Moreover, as $R^2$ of exploration innovation ($R^2 = 0.318$) is larger than $R^2$ of exploitation ($R^2 = 0.277$), it is possible to claim that ICs of Mozambican SMEs are exerting more influence on creating innovative activities and adopting innovative strategies to support radical new products and markets to cope with changing (international) environments than in embracing innovative technologies to improve their existing product portfolio associated with the markets they are actually serving. As such, complementing what was previously found [26,35,56], it is possible to claim that Mozambican SMEs use their ICs to address risky innovative strategies than to address incremental innovations for the markets they need. Due to the limited quantities of resources and capabilities to develop their innovation capabilities base, one possible explanation for that is the demand of ICs to be tuned to international more demanding markets and investing in riskier more innovative products and services is larger than the demand of ICs to satisfy the demand for products and services that are usually associated with markets which the company already serves.

When analyzing exploitation and exploration innovation as antecedents of export performance, it is possible to conclude that exploitation innovation ($\beta = 0.397$; $p < 0.001$) is more intense than exploration innovation ($\beta = 0.224$; $p < 0.001$), which indicates that short-term, incremental innovative activities are more prevalent than long-term, more risky innovation activities when explaining export performance. This result confirms the results obtained by Ribau et al. [26]. As such, although it is possible to claim that Mozambican SMEs manage to compete abroad with both exploration and exploitation innovation capabilities, the latter are the ones that influence export performance the most. This means that Mozambican SMEs depend more on incremental, short-term capabilities, which may indicate that, despite their ambidexterity, they do not manage to take full advantages of radical, long-term innovation activities when compared to incremental, short-term innovation activities, which may be the consequence of resource limitations, limited access to finance and international markets, limiting their capacity to reconfigure their existing resources and capabilities and, thus, limiting their international competitiveness.

The results indicate that the mediating effects of exploration and exploitation innovation capabilities are important, although the mediating effect of exploitation innovation is slightly higher. This result confirms previous literature defending that the mediating effect of exploitation innovation has a positive impact on export performance [26,52,55,56], which indicates that Mozambican SMEs seek to enhance their internal competencies, contributing positively to export performance, although they have a stronger short-term orientation *vis-à-vis* their long-term orientation.

Likewise, the specific indirect relationships between innovation capabilities and export performance, mediated by exploration innovation, are also important. Thus, we can state that Mozambican exporting SMEs seek to take advantage of the long-term capabilities of exploration innovation, to generate new products and services characterized by unique skills and capabilities, which are riskier than those needed to compete based on short-term orientation. This exploration innovation means long-term experimentation with new alternatives, where economic returns are, of course, uncertain. Therefore, the exploration of new knowledge and skills in a long-term perspective also stimulates and helps to increase the export performance of Mozambican SMEs.

If ICs are important for firms to compete internationally, it is also important to claim that Mozambican SMEs, as an example of an emerging country, have managed to deploy their ambidexterity to compete in international markets. If both exploration and exploitation innovation capabilities are important, because they represent different types of capabilities and different types of short- and long-term perspectives, it is clear that the trade-off between both is not easily handled. The resource constraints in emerging markets, as is the case of Mozambique, make exploitation innovation capabilities more prevalent than exploration innovation capabilities as Mozambican SMEs are not very familiar competing in highly demanding markets. As such, despite all the efforts they go through when they try to serve highly demanding international markets, exploitation capabilities stand out *vis-à-vis* exploration capabilities as it is easier for them to serve stable markets that need incremental, short-term innovation activities than highly innovative markets that seek radical, long-term innovation activities. Moreover, although the focus of the analysis has been on the resources, it is possible to claim that the lack of competitive internal market, as well as in many emerging economies, jeopardizes the proper evolution to more demanding risk-oriented radical innovation perspectives that those Mozambican SMEs are likely to find in international markets.

## 6. Conclusions

This paper assesses the impact of innovation capabilities on the export performance of 250 Mozambican SMEs and the mediation effects of exploration and exploitative innovation. The results indicate that ICs have a significant direct effect on the export performance of these SMEs. However, it is also found that Mozambican SMEs use existing knowledge and skills—exploitation innovation—and seek new knowledge and skills in developing new products and processes—exploration innovation—to increase their performance in the international market. Thus, we can state that the exploration and exploitation effects are important in the relationship between innovation capabilities and export performance.

This study has broad implications at several levels. First, it contributes to the literature on SME internationalization by addressing the importance of innovation skills and the mediating effects of exploration and exploitation innovation on the relationship between ICs and export performance, especially in less developed contexts, such as Mozambique. As can be seen from the results, the indirect effects complement the direct effects.

The second implication is related to the use of a questionnaire on innovation skills in emerging economies, namely in exporting Mozambican firms, which helps to understand that innovation skills are generic to various countries and are not necessarily confined to firms in developed countries. Thus, it can be stated that firms in emerging countries can and should make use of innovation capabilities from a holistic perspective to implement production, marketing, R&D, learning, organizational, resource exploitation, and strategic capabilities to generate their new products and services to compete in international markets. These capabilities are broader, based on exploration and exploitation perspectives, i.e., from the short- and long-term perspectives. If these seven capabilities are proven to add value to enhance the competitiveness of Mozambican SMEs and their export performance, future studies should explore how these Mozambican SMEs interrelate their core innovation capabilities (production, marketing, and R&D) and supplementary capabilities (learning, organizational, resource exploitation, and strategic).

A third implication is related to the ambidexterity of Mozambican international SMEs. As found in this paper, both exploitation and exploration capabilities are present as mediators of the relationship between innovation capabilities and export performance, although short-term oriented exploitative capabilities are more prevalent than long-term exploration capabilities. As such, it can be concluded that Mozambican firms need to nurture their long-term oriented perspective in order to adopt entirely new products and services while continuously improving their actual resources and processes. To do this, they need to balance their risk-taking behavior in choosing to solve their incremental innovation and balance their abilities to generate long-term disruptive innovation.

In addition to studying the relationship between ICs and export performance, further research should address the effects of the intensity, strategy, and competitive advantages of SMEs in emerging countries.

One of the limitations of this study is the fact that cross-sectional research was used, without taking into consideration the intrinsic characteristics of a longitudinal research. The use of only one informant per firm is another limitation.

**Author Contributions:** Conceptualization, A.C.M., E.C.N. and C.R.; methodology, A.C.M., E.C.N. and C.R.; validation, A.C.M., E.C.N. and C.R.; formal analysis, A.C.M., E.C.N. and C.R.; investigation, E.C.N.; data curation, A.C.M. and E.C.N.; writing—original draft preparation, E.C.N.; writing—review and editing, A.C.M., E.C.N. and C.R.; supervision, A.C.M. and C.R. All authors have read and agreed to the published version of the manuscript.

**Funding:** This research received no external funding.

**Institutional Review Board Statement:** The study was conducted in accordance with the Declaration of Helsinki, and approved by Universidade Zambeze.

**Informed Consent Statement:** Informed consent was obtained from all subjects involved in the study.

**Data Availability Statement:** Not applicable.

**Conflicts of Interest:** The authors declare no conflict of interest.

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
