# Peer review of "The Importance of Exploration and Exploitation Innovation in Emerging Economies"

_2199-8531, doi:10.3390/joitmc8030140_

Round 1

Reviewer 1 Report

1. The abstract should contain purpose, methodology, fingings, and value. Please revise.

2. The state of the art needs serious improvement, the introduction section lack literature and hence it’s not informative in terms of past results, the need for the current study, its future, etc.

3. Based on Section 2, the authors did not hypothesize the direct effect between innovation capabilities and exploration/exploitation innovation as well as between exploration/exploitation innovation and export performance. However, Figure  shows these direct effect in solid lines.

4. Correct the heading of Table 5.

5. The authors should add the discussion section and explain on how the findings contribute to theory and practice.

Author Response

Many thanks for your comments and for helping improve the document.

Best regards

Reviewer 2 Report

The authors have made a decent attempt at exploring how exploration and exploitation innovation influence export performance in the context of emerging economies. In order to contribute to the improvement of the paper, the following points deserve attention from the authors. 

1)    The introduction does not count with the contributionHighlight the theoretical and practical contributions of the paper in the introduction section.

2) There is no difference between hypotheses 2a and 2b. The authors should change "exploration" to "exploitation" in hypothesis 2b.

3) What is the profile of the respondents? Which sector each one of them works in? Position held? Years of experience?

4) What is the source and rationale for using each variable under each construct? 

5) The discussion section is weak. The discussion section can be split based on each finding. I suggest connecting each finding with its respective theoretical and practical contribution.

Author Response

(The authors gave the same response as above.)

Reviewer 3 Report

Would you please address a little about how the situation in Mozambique matters in the results?

Author Response

(The authors gave the same response as above.)

Round 2

Reviewer 1 Report

This version of manuscript is acceptable for publication.

Author Response

Many thanks for your helping us improve the article during the process and thanks for the acceptance of the article.

Best regards

Reviewer 2 Report

The paper can be accepted in its present form. 

Author Response

(The authors gave the same response as above.)
